# Application of Deep Learning in Image Recognition of Citrus Pests

Xinyu Jia [1,2,†], Xueqin Jiang [1,2,†], Zhiyong Li [1,2,*], Jiong Mu [1,2], Yuchao Wang [3] and Yupeng Niu [1,2]

1   College of Information Engineering, Sichuan Agricultural University, Ya'an 625000, China; jiaxinyu@stu.sicau.edu.cn (X.J.); jiangxueqin@stu.sicau.edu.cn (X.J.); jmu@sicau.edu.cn (J.M.); 202105647@stu.sicau.edu.cn (Y.N.)
2   Ya'an Digital Agricultural Engineering Technology Research Center, Ya'an 625000, China
3   College of Mechanical and Electrical Engineering, Sichuan Agricultural University, Ya'an 625000, China; wangyc0918@yahoo.co.jp
*   Correspondence: lzy@sicau.edu.cn; Tel.: +86-138-8221-3811
†   These authors contributed equally to this work.

**Abstract:** The occurrence of pests at high frequencies has been identified as a major cause of reduced citrus yields, and early detection and prevention are of great significance to pest control. At present, studies related to citrus pest identification using deep learning suffer from unbalanced sample sizes between data set classes, which may cause slow convergence of network models and low identification accuracy. To address the above problems, this study built a dataset including 5182 pest images in 14 categories. Firstly, we expanded the dataset to 21,000 images by using the Attentive Recurrent Generative Adversarial Network (AR-GAN) data augmentation technique, then we built Visual Geometry Group Network (VGG), Residual Neural Network (ResNet) and MobileNet citrus pest recognition models by using transfer learning, and finally, we introduced an appropriate attention mechanism according to the model characteristics to enhance the ability of the three models to operate effectively in complex, real environments with greater emphasis placed on incorporating the deep features of the pests themselves. The results showed that the average recognition accuracy of the three models reached 93.65%, the average precision reached 93.82%, the average recall reached 93.65%, and the average F1-score reached 93.62%. The integrated application of data augmentation, transfer learning and attention mechanisms in the research can significantly enhance the model's ability to classify citrus pests while saving training cost and time, which can be a reference for researchers in the industry or other fields.

**Keywords:** citrus; pest; classification; dataset; transfer learning; AR-GAN; attention mechanism

## 1. Introduction

Citrus is considered to be among the most commonly grown fruits throughout the world and is an essential economic pillar for farmers at home and abroad [1,2]. The daily management of citrus orchards mainly relies on the old management model, which relies on pesticides to control pests in the orchard and seriously destroys the ecological balance between species in citrus orchards [3–6]. As computer vision and image processing technology continues to advance at an accelerated pace, Resnet50, VGG16 and MobileNetV2 have been widely used in smart agriculture [7–13]. Hamid et al. [9] used MobileNetV2 to classify seeds and achieved 95% accuracy. Farhana et al. [14] used a two-stage Convolutional Neural Network (CNN) to detect and classify citrus diseases (a total of 598 sample images, three disease samples and healthy samples) with a final accuracy of 95.8%. Zhou et al. [15] proposed a classification recognition method combining InceptionV3 with transfer learning and conducted experiments on their own collected citrus pest dataset with the accuracy of 96.81%. Ramazan et al. [10] used transfer learning combined with CNNs to experiment on their own dataset (1519 sample images) using four pretrained

CNNs, where AlexNet achieved the highest accuracy of 99.33%. Therefore, the use of CNNs for automatic identification of citrus pests becomes a feasible and low-cost method.

Generative Adversarial Network (GAN) data augmentation methods emerged because classical data augmentation methods cannot replicate the greater variability in various natural conditions. GAN networks can generate images that visually appear more naturally diverse over different domains [16–18]. Ding et al. [19] introduced re- cursive Attentive Recurrent Generative Adversarial Networks (AR-GAN) as a means of improving image augmentation, and the method showed superior performance on four public datasets. Haseeb et al. [20] improved the AR-GAN data augmentation network by optimizing the loss function, resulting in an obvious increase in accuracy (+5.2%). Xiao et al. [21] conducted experiments on eight classification networks, including Residual Neural Network (ResNet) 50, ResNet101, Dense Convolutional Network (DenseNet) 121, and MobileNetV3, with datasets from the publicly available dataset PlantVillage and self-collected images of citrus disease samples (4516 images, 6 disease categories); introducing Texture Reconstruction Loss CycleGAN to increase sample size resulted in an average classification accuracy improvement from 92.06% to 94.82% (+2.76%).

Training a large CNN from scratch is a time-consuming process that requires a significant number of iterations, and the application of transfer learning can effectively solve this problem when the training dataset is too small and the target domain data is difficult to collect [22]. Aggarwal et al. [7] used VGG16, DenseNet169 and ResNet152 for fine-tuned parameter-based transfer learning training on the "Human Protein Atlas Image Classification" dataset. Gulzar et al. [8] used the improved model TL-MobileNetV2 combined with transfer learning to achieve 99% accuracy on the fruit dataset (26,149 images, 40 species). Mamat et al. [23] combined transfer learning on the YOLO model to classify the ripeness of oil palm fruits with a final mAP of 99.5%. Yang et al. [24] combined the VGG16 network introducing transfer learning with CNN13 on their self-collected citrus pest and disease dataset (2071 sample images, 4 pests and 1 healthy sample image) and proposed a novel joint network OplusVNet, and the final model predicted the results with 99.00% classification accuracy.

Humans can naturally find salient regions in complex scenes, and influenced by this observation, computer vision has introduced attention mechanisms to mimic the human visual system to focus on certain features that require special attention [25]. Bollis et al. [26] conducted experiments on their self-collected citrus pest dataset (a total of 10,816 samples, 6 classes of mite pests) and the CNN achieved 92.4% accuracy after adding the attention mechanism. Yang et al. [27] suggested a CNN capable of focusing on both channel and spatial information for the classification task of pests, and the method achieved 73.29% accuracy on the IP102 dataset. Dai et al. [28] embedded a Convolutional Block Attention Module (CBAM) on a High Reduction Network (HRNet) to focus on small size features for the task of citrus pest identification and obtained an accuracy of 88.78% in the end.

In this paper, a citrus pest dataset was collated and named IP_CitrusPests. We then expanded the datasets to 1500 pests per category by the AR-GAN [20] data augmentation technique. Then we introduced transfer learning to fine-tune the parameters on the models. Finally, we added the appropriate attention mechanisms on each of the three classification networks to attain precise image categorization within the domain of citrus pests. This study applies CNN to the application of citrus pest image recognition, which can accumulate a certain foundation for the subsequent development of citrus pest recognition technology. The specific research objectives were as follows:

- We will reorganize a citrus pest dataset IP_CitrusPests based on the IP102 [29] citrus pest dataset by crawling and filtering, which will include larvae and adults of 14 classes of citrus pests.
- Considering the training time and computational resources, we will choose the three most popular networks in the image classification field, Resnet50, VGG16 and MobilenetV2, to identify 14 classes of citrus pests in the IP_CitrusPests dataset.

- Considering the difficulty of collecting individual pest samples, which makes the IP_CitrusPests dataset of this study have the problem of inter-class imbalance, we will expand the datasets to 21,000 images by the AR-GAN [20] data augmentation technique, in which the number of each class of pests can reach 1500.
- In order to reduce the number of parameters for model training and reduce the computational cost, this study will introduce transfer learning by fine-tuning the parameters of the pre-trained weights trained on the ImageNet public dataset [30] on three classification network models.
- In order to improve the ability of the model to focus on the pest itself in a complex context, this study will add corresponding attention mechanisms to each of the three models according to their differences, including an attempt to add the Relation-Aware Global Attention (RGA) [31], which can consider global information from spatial and channel dimensions, to the appropriate position in ResNet50; an attempt to add the Efficient Channel Attention (ECA) [32], which can extract inter-channel dependencies, to VGG16; and an attempt to embed the Coordinate Attention (CA) [33], which can consider the relationship between space, channel and location simultaneously, into MobileNetV2.

## 2. Literature Review

CNNs require training samples under different conditions such as different scenes, orientations, locations and luminance to enhance the network's robustness. Studies have shown that the existing citrus pest datasets are generally small in number or not comprehensive in species coverage. Hafiz et al. [34] used a dataset including only 759 citrus sample images to train the network. Rahman et al. [14] collected their own citrus pest dataset including only 5 classes of pests with a total of 506 sample images. Chen et al. [35] acquired 950 fruit images for citrus ripeness detection. Wang et al. [12] collected 2 types of citrus pests with the dataset of only 1000 sample images. In general, most of the existing open-source citrus pest datasets have problems such as too few samples or insufficient samples containing complex environments, and the size and quality of the datasets are key issues in how to maximize the application of deep learning techniques in practical smart agriculture applications [36].

A comprehensive dataset, IP102, was gathered by Wu et al. with a particular focus on pest recognition [29], which included 102 pest species with as many as 75,000 images but still suffers from a high imbalance in the number of samples; they used AlexNet, GoogleNet, VGG16 and ResNet50 on the IP102 dataset to perform classification experiments, in which the highest accuracy of 49.4% was achieved on ResNet50. The current manual feature extraction methods and deep learning feature extraction methods are not good enough to learn the complex features of the samples in IP102. Thus, the aim of this study is to create a dataset tailored to citrus pest investigation, utilizing the IP102 dataset as a foundation. Moreover, this paper will address the issue of imbalanced datasets in this context.

Most of the citrus pest datasets have limited shooting scenes and conditions, and the quantity of pictures cannot reach the scale required for network training, so we need to train the CNNs by additional synthetic data. Morteza et al. [37] used traditional data enhancement (rotation, mirroring and adding Gaussian noise to RGB images) to expand the citrus pest dataset for a self-collected dataset (total of 3 pests, 1774 sample images), ultimately achieving 99.04% accuracy. You et al. [13] conducted experiments on a self-collected citrus pest dataset (total of 16,258 images, including 8 classes of pests, 5 common citrus diseases and 1 healthy leaf) were experimented on, and the dataset after performing traditional data enhancement (scaling, rotation, panning, etc.) performed best in the DenseNet classification network with an accuracy of 0.937. However, the modification of images obtained using classical data augmentation resulted in relatively small changes in the images, and the GAN data augmentation generates samples that introduce more variability and can further enrich the dataset. Therefore, we will further expand the dataset of this study using AR-GAN data augmentation to address the imbalance of the dataset.

## 3. Materials and Methods

### 3.1. Dataset Collection

This study re-collected a citrus pest dataset, IP_CitrusPests, for citrus pest identification based on the IP102 [29]. The IP_CitrusPests was assembled by means of image collection, followed by image screening and annotation. During the phase of collecting images, we chose to use web crawlers to collect a wide range of 14 types of citrus pest samples and then filtered and organized the results from the web crawlers to make the data samples include more complex environments and backgrounds as much as possible to better match the real application scenarios, as shown in Figure 1. During the phase of annotating data, an agricultural expert in citrus research was invited to annotate our screened sample images with the original IP102 sample data.

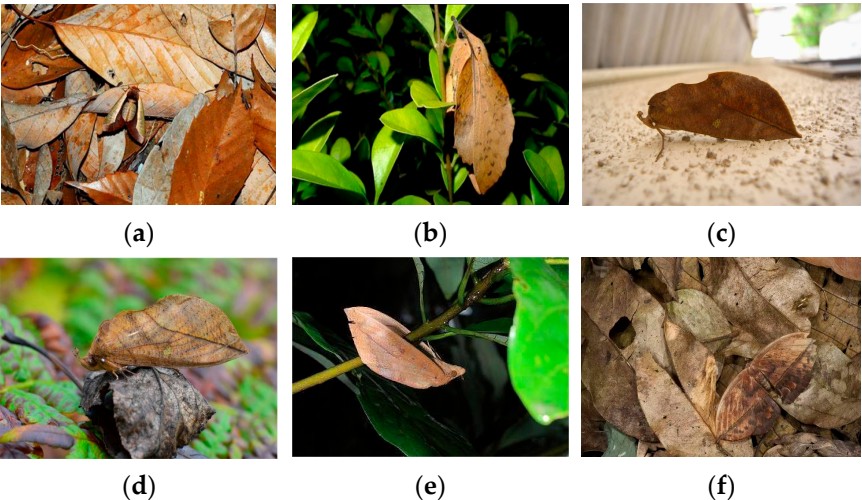

(a)      (b)      (c)

(d)      (e)      (f)

**Figure 1.** Example of individual samples with complex backgrounds from Adris Tyrannus in the IP_CitrusPests dataset. (**a**) refers to Adris Tyrannus in a complex leaf background, (**b**) refers to Adris Tyrannus in a natural leaf background, (**c**) refers to Adris Tyrannus in a normal ground background, (**d**) refers to Adris Tyrannus in a wilted leaf background, (**e**) refers to Adris Tyrannus in a natural environmental background, (**f**) refers to Adris Tyrannus in a more complex leaf background.

Table 1 illustrates the IP_CitrusPests dataset proposed in this study comprising a grand total of 5182 sample images, covering a total of 14 citrus pests, including *Adris Tyrannus*, *Aleurocanthus spiniferus*, *Bactrocera tsuneonis*, *Ceroplastes rubens*, *Chrysomphalus aonidum*, *Dacus dorsalis Hendel*, *Panonchus citri McGregor*, *Papilio xuthus*, *Parlatoria zizyphus Lucus*, *Phyllocoptes oleiverus ashmead*, *Prodenia litura*, *Toxoptera citricidus*, *Toxoptera aurantii* and *Unaspis yanonensis*. The sample types cover almost all pest species that are susceptible to affect citrus plants, and this dataset is more diversified than previous datasets in related fields. Table 1 shows that the largest number of samples was 1529 for the *Prodenia litura*, while the smallest number was 74 for the *Parlatoria zizyphus Lucus*, and the dataset has a high imbalance in the number between categories. The dataset was partitioned into three subsets, namely the training set, validation set, and test set, in a 6:2:2 ratio. For the classification network, 3110 sample images from the IP_CitrusPests dataset were allocated for training, while 1036 sample images each were assigned for validation and testing.

**Table 1.** Sample size statistics for the IP_CitrusPests dataset.

| Serial Number | Pest Name | Number of Pest Samples |
|:---:|:---:|:---:|
| 1 | *Adris tyrannus* | 415 |
| 2 | *Aleurocanthus spiniferus* | 502 |
| 3 | *Bactrocera tsuneonis* | 127 |
| 4 | *Ceroplastes rubens* | 256 |
| 5 | *Chrysomphalus aonidum* | 193 |
| 6 | *Dacus dorsalis Hendel* | 377 |
| 7 | *Panonchus citri McGregor* | 433 |
| 8 | *Papilio xuthus* | 449 |
| 9 | *Parlatoria zizyphus Lucus* | 74 |
| 10 | *Phyllocoptes oleiverus ashmead* | 167 |
| 11 | *Prodenia litura* | 1529 |
| 12 | *Toxoptera citricidus* | 143 |
| 13 | *Toxoptera aurantii* | 146 |
| 14 | *Unaspis yanonensis* | 371 |
| Total | | 5182 |

### 3.2. AR-GAN Data Augmentation

CNNs require a large amount of sample data for training. Collecting adequate and varied data to accomplish the task is challenging due to the intricacy of the real-world citrus growth environment and field climate. Therefore, it becomes necessary to incorporate data augmentation to increase the quantity and diversity of training sets and enhance the model's resilience. In order to tackle the issue of sample imbalance present in the IP_CitrusPests, especially for individual pest categories with too few samples such as *Parlatoria zizyphus Lucus*, *Phyllocoptes oleiverus ashmead* and *Toxoptera citricidus*, we used AR-GAN data augmentation [20] to increase their sample size, and finally, the sample size of each pest category in the IP_CitrusPests dataset reached 1500.

AR-GAN is a GAN data augmentation method suitable for use in unsupervised image translation environments. AR-GAN optimizes Attentive Recurrent Loss (ARL) [38] based on CycleGAN [39] and DualGAN [40], which strengthens the similarity between real and generated images to make the dataset sample environment and background richer and more diverse. Figure 2 illustrates the complete structure of AR-GAN, with A and B denoting two distinct categories of citrus pests, and the generators ($G_{AB}$, $G_{BA}$) combine the pest features of the B (A) sample during the training process, and after multiple residual blocks, transfers the features of one class of pest sample images to another domain to generate a fake A (B) class sample. To determine if the generated image belongs to a specific domain, the discriminator ($D_A$, $D_B$) is dedicated to each domain. The generator and discriminator perform forward propagation and backward propagation, respectively, to continuously optimize their parameters by adversarial means to generate realistic images and judge the authenticity. The data and parameters are iteratively passed between A→B→A (green) and B→A→B (red).

### 3.3. Transfer Learning

Transfer learning was initially developed to accelerate the training of target domain models by utilizing existing labeled data to transfer knowledge to un-labeled data domains, resulting in more efficient and accurate model training while reducing time and cost [41]. By transferring the general knowledge contained in various but interconnected source domains, the accuracy of the CNN in the target domain can be enhanced [42].

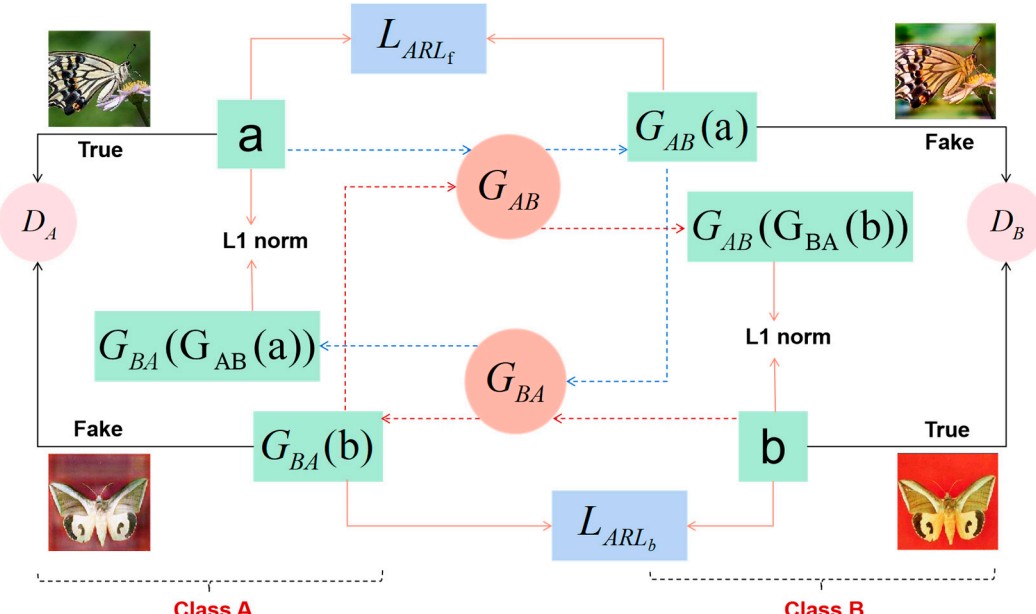

**Figure 2.** Overall framework of AR-GAN.

Based on the fact that the dataset IP_CitrusPests processing domain has the same feature space as the large public dataset, the parameter-based isomorphic transfer learning method was implemented to fine-tune the replacement of the final classification layer in this research. Based on the specifics of the IP_CitrusPests dataset, we chose to introduce pretraining weights on the ImageNet [30] dataset, which is the most widely used and has the greatest impact compared with the IP_CitrusPests dataset; the ImageNet dataset is large enough and the data samples are diverse and pluralistic enough.

Therefore, in this experiment, the network parameters trained on ImageNet were utilized to transfer knowledge to the classification network trained on IP_CitrusPests, and the network model replacing the new layer is trained, updated and iterated. The ImageNet dataset has 1000 categories making the last layer of the CNN 1000 neurons, while the dataset IP_CitrusPests used in this study has 14 different pest categories, so the last layer of the model in this study should have 14 neurons. After replacing the new layer, the classification network is frozen (untrainable layer) for all layers except the last layer (trainable layer), and only the weight of the final layer is updated. Table 2 demonstrates that all three models after parameter fine-tuning largely reduce the quantity of parameters used for training and lower the computational cost.

**Table 2.** Relevant parameters (input shape, convolution layer, full connected layer, no-trainable parameter, trainable parameter and total parameter) of the three models (ResNet50, VGG16 and MobileNetV2) after parameter fine-tuning.

| Model | Input Shape | Convolution Layer | Fully Connected Layer | No-Trainable Parameter | Trainable Parameter | Total Parameter |
|---|---|---|---|---|---|---|
| Resnet50 | (224, 224, 3) | 49 | 1 | 23,508,032 | 28,686 | 23,536,718 |
| VGG16 | (224, 224, 3) | 13 | 3 | 134,260,544 | 57,358 | 134,317,902 |
| MobileNetV2 | (224, 224, 3) | 52 | 1 | 2,223,872 | 17,934 | 2,241,806 |

### 3.4. Attention Mechanism

The attention mechanism is designed to concentrate on specific details within an image, locating the relevant features in the region of interest while suppressing irrelevant data. In order to accurately locate the pest itself in the context of complex environments, this study addresses the differences between the ResNet50, VGG16 and MobileNetV2

classification network models and introduces attention mechanisms corresponding to each of these networks.

### 3.4.1. ResNet_RGA

Relation-Aware Global Attention (RGA) [31] is applied to consider the feature at each position in the feature graph as a node and to global-scale relevance and semantic information by emphasizing symmetric relationships among the nodes. This attention module acts on both spatial and channel dimensions and achieves high performance in multiple datasets with a limited number of growing parameters. For the ResNet50 classification model, by simultaneously considering the global relational information, we applied the unique RGA module to the ResNet50 network to obtain the feature vector representation of the images. As shown in Figure 3, we added the RGA module after res_layer1, res_layer2, res_layer3 and res_layer4 of ResNet50. Notably, the last spatial down-sampling operation in res_layer4 of ResNet50 was removed, and we set its step stride to 1.

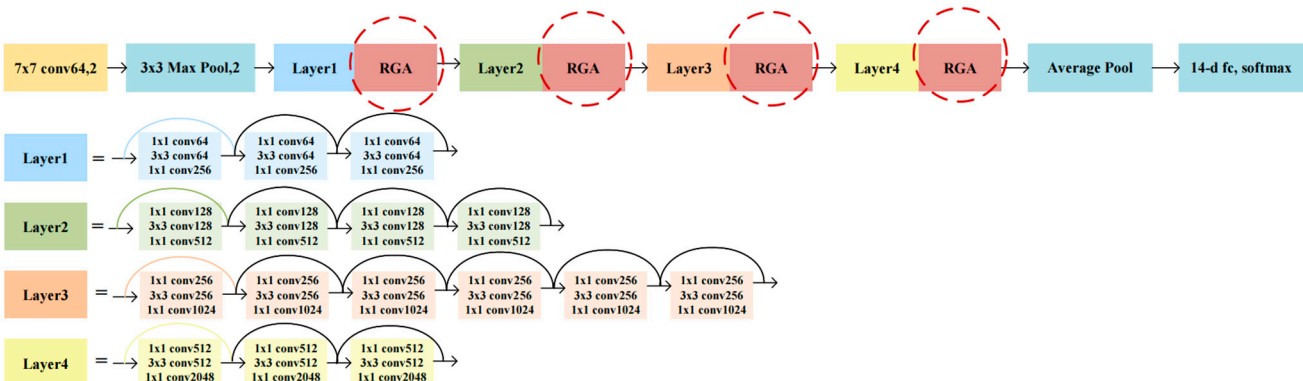

**Figure 3.** Structure of ResNet50_RGA and red circle represented the newly embedded RGA module.

### 3.4.2. VGG_ECA

Efficient Channel Attention (ECA) [32] employs $1 \times 1$ convolution to extract information across various channels, which enables local cross-channel interaction, extracts inter-channel dependencies, and this method prevents the reduction of channel dimensionality while learning the attention information for each channel. In this study, the ECA attention mechanism model was added at the end of VGG16 (after layer 5), as shown in Figure 4, to strengthen the overall channel characteristics and improve the model performance while reducing the amount of network parameters.

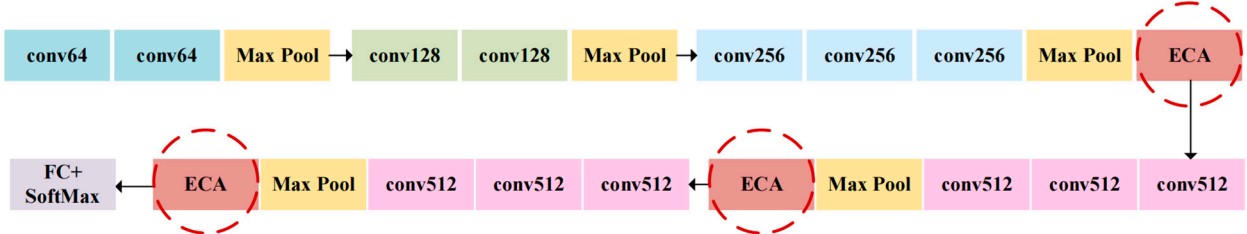

**Figure 4.** Architecture of VGG16_ECA and red circle represented the newly embedded ECA module.

### 3.4.3. MobileNetV2_CA

Coordinate Attention (CA) [33] is a module that embeds position into channel information, which considers not only the relationship between space and channel but also the long-range dependency problem. CA decomposes channel attention into two independent one-dimensional features encoding procedures, which gather features along two different spatial orientations so that on the one hand this allows the module to capture long-distance

dependencies along one direction while retaining accurate position information along the other spatial orientation. Thus, it acts more complementarily on the input feature map and enhances the attention of the network to the parts that should be attended to, and CA is flexible and lightweight enough to be simply inserted into lightweight networks.

For MobileNetV2, this study added the CA to the last reverse residual module of the network (shown in Figure 5), which enables the model to prioritize significant features in both the channel and spatial dimensions, leveraging the captured location information to accurately identify regions of interest.

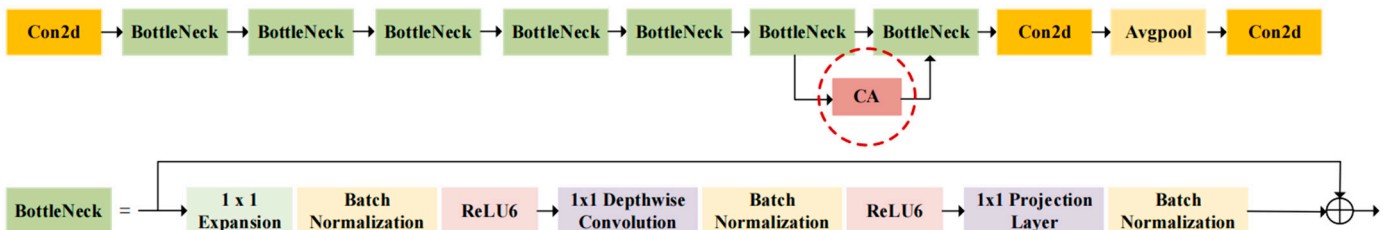

**Figure 5.** Structure of MobileNetV2_CA and red circle represented the newly embedded CA module.

### 3.5. Evaluation Indicators

For the performance of IP_CitrusPests on three different classification network models, to evaluate the performance of the model, we utilized various evaluation metrics such as accuracy, precision, recall and F1-score. Accuracy is a widely used performance metric in classification tasks, representing the ratio of correctly classified samples to the overall count of samples. Precision is the proportion of samples with positive prediction results of the true cases. Recall is the ratio of samples with positive prediction results among those with true positive cases, and this metric can show the capacity of the model to discriminate between positive samples. F1-score is a composite metric that considers both precision and recall of the prediction results.

The expressions for accuracy, precision, recall, and F1-score are as follows:

$$\text{Accuracy} = \frac{\text{TP} + \text{TN}}{\text{TP} + \text{TN} + \text{FP} + \text{FN}} \tag{1}$$

$$\text{Precision} = \frac{\text{TP}}{\text{TP} + \text{FP}} \tag{2}$$

$$\text{Recall} = \frac{\text{TP}}{\text{TP} + \text{FN}} \tag{3}$$

$$\text{F1} - \text{score} = \frac{2 \times \text{Precision} \times \text{Recall}}{\text{Precison} + \text{Recall}} \tag{4}$$

where TP denotes the count of positive samples that are predicted as positive, while TN represents the count of negative samples that are predicted as negative. Similarly, FP indicates the count of negative samples that are predicted as positive, and FN indicates the count of positive samples that are predicted as negative.

### 3.6. Training Environment

In this experiment, the AR-GAN model was based on PyTorch 1.12.1, and the ResNet50, VGG16 and MobileNetV2 networks were also trained on PyTorch 1.12.1 framework. The computational unified device architectures were CUDA 11.6 (From graphics card manufacturer NVIDIA, California, USA) and python 3.8. The experiments were conducted on a system with Ubuntu 20.04.5 LTS (Developed by Canonical, London) equipped with a processor that has a base frequency of 2.30 GHz and 128 GB RAM. The graphics card used in this experiment was NVIDIA Quadro RTX 5000 (From graphics card manufacturer

NVIDIA, California, USA). Details regarding the specific values of each hyperparameter used in the experiment are presented in Table 3.

**Table 3.** This is the model parameter in the experiment.

| Parameter | AR-GAN | ResNet50, VGG16 and MobileNetV2 (Freeze Process) | ResNet50, VGG16 and MobileNetV2 (Unfreeze Process) |
|---|---|---|---|
| Batch Size | 8 | 4 | 8 |
| Epoch | 100,000 | 100 | 100 |
| Optimizer | Adam | Adam | Adam |
| Learning Rate | 0.0001 | 0.001 | 0.0001 |

## 4. Results

### 4.1. AR-GAN Data Augmentation Evaluation

In this experiment, the AR-GAN was used to expand the samples of the IP_CitrusPests. The number of samples for all 14 pest categories was expanded to 1500, and the overall number of the dataset reached was 21,000.

Some samples of the AR-GAN data-enhanced dataset are shown in Figure 6. Each of which learns the sample features of that pest category. It is obvious from Figure 6 that the data features of the sample images after AR-GAN data augmentation are more complex and can offer more valuable feature information.

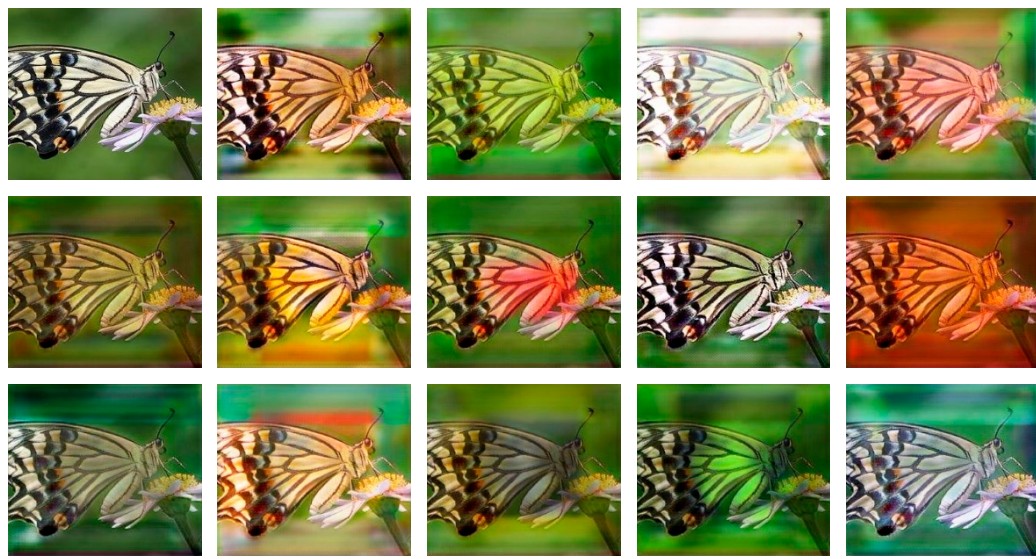

**Figure 6.** AR-GAN data augmentation effect. The top-left image in the figure represents the primitive image, while the others are samples that were generated using the AR-GAN network.

In Table 4, the classification accuracies of ResNet50, VGG16 and MobileNetV2 are compared for each pest class before and after the application of AR-GAN, and the results demonstrated a noticeable improvement in the recognition accuracy of each pest category. In particular, pest species with relatively small sample sizes before data augmentation in the IP_CitrusPests dataset, such as *Bactrocera tsuneonis*, *Parlatoria zizyphus Lucus* and *Phyllocoptes oleiverus ashmead*, had low classification accuracy on models.

**Table 4.** Comparison of the recognition accuracy of each pest category on the ResNet, VGGNet and MobileNet for 14 categories of pests before and after the AR-GAN data augmentation technique.

| Model | AR-GAN | AT [1] | As [2] | Bt [3] | Cr [4] | Ca [5] | DdH [6] | PcM [7] | Px [8] | PzL [9] | Poa [10] | Pl [11] | Tc [12] | Ta [13] | Uy [14] |
|---|---|---|---|---|---|---|---|---|---|---|---|---|---|---|---|
| Resnet50 | No | 75.00 | 89.33 | 31.58 | 67.11 | 61.40 | 78.76 | 82.95 | 82.84 | 54.55 | 68.00 | 83.19 | 44.19 | 42.86 | 72.07 |
|  | Yes | 90.00 | 85.78 | 63.11 | 71.56 | 81.11 | 88.44 | 79.56 | 95.33 | 87.78 | 94.89 | 91.33 | 72.44 | 71.33 | 72.89 |
| VGG16 | No | 62.10 | 70.00 | 0.00 | 7.89 | 1.75 | 54.87 | 71.32 | 81.34 | 0.00 | 0.00 | 73.80 | 6.98 | 0.00 | 56.76 |
|  | Yes | 82.89 | 86.00 | 75.11 | 77.78 | 92.89 | 87.11 | 82.67 | 92.00 | 94.67 | 90.22 | 72.00 | 75.33 | 84.00 | 72.22 |
| MobileNetV2 | No | 56.45 | 80.67 | 13.16 | 44.74 | 43.86 | 69.91 | 67.44 | 77.61 | 59.09 | 42.00 | 78.82 | 23.26 | 16.67 | 64.86 |
|  | Yes | 75.78 | 84.44 | 46.22 | 53.56 | 65.11 | 75.56 | 72.00 | 91.33 | 86.67 | 89.33 | 88.00 | 67.56 | 52.89 | 59.78 |

[1] AT refers to Adris Tyrannus. [2] As refers to Aleurocanthus spiniferus. [3] Bt refers to Bactrocera tsuneonis. [4] Cr refers to Ceroplastes rubens. [5] Ca refers to Chrysomphalus aonidum. [6] DdH refers to Dacus dorsalis Hendel. [7] PcM refers to Panonchus citri McGregor. [8] Px refers to Papilio xuthus. [9] PzL refers to Parlatoria zizyphus Lucus. [10] Poa refers to Phyllocoptes oleiverus ashmead. [11] Pl refers to Prodenia litura. [12] Tc refers to Toxoptera citricidus. [13] Ta refers to Toxoptera aurantii. [14] Uy refers to Unaspis yanonensis.

According to the results presented in Table 4, these pest categories with small sample sizes were significantly enhanced by AR-GAN data. In the ResNet50 classification model, the accuracy of *Bactrocera tsuneonis*, which had only 127 samples before data augmentation, was 31.58%, but after AR-GAN was expanded to 1500 samples the accuracy of this pest reached 63.11%, an improvement of 31.53%. Additionally, the accuracy of *Parlatoria zizyphus Lucus*, which had only 74 samples before data augmentation, was 54.55%, but after AR-GAN was expanded to 1500 samples the accuracy of this pest reached 87.78%, an improvement of 33.23%. For the VGG16 model, the number of samples of *Toxoptera citricidus* was expanded from 143 to 1500 after AR-GAN, and the accuracy also improved from 6.98% to 75.33%, with an overall improvement of 68.35%. For the MobileNetV2 model, the number of samples of *Toxoptera aurantii* was expanded from 146 to 1500 after data augmentation, and the accuracy also showed a significant improvement from 16.67% to 52.89%, an improvement of 36.22%.

Table 5 presents a comparison of evaluation metrics for the IP_CitrusPests dataset, which was expanded using the AR-GAN on ResNet50, VGG16 and MobileNetV2. Overall, with the experimental setup of balancing the number of 14 classes of pest samples by introducing only the AR-GAN method, all three classification networks showed substantial improvement, with the average accuracy increasing from 65.52% to 79.02%, a total improvement of 13.50%. Among them, the VGG16 classification network had the largest before-and-after accuracy improvement, Accuracy directly improved from 55.33% to 83.21%, a total improvement of 27.88%, while ResNet50 and MobileNetV2 improved by 5.81% and 6.80%, respectively. This experiment fully demonstrated that the AR-GAN data augmentation can cope well with the highly unbalanced dataset and make the samples more complex and diverse, thus improving the robustness of the network and enabling the CNN to better cope with real and complex environments.

**Table 5.** The comparison of the precision, recall, F1-score and accuracy of the IP_CitrusPests dataset expanded by AR-GAN on ResNet50, VGG16 and MobileNetV2.

| Model | AR-GAN | Precision (%) | Recall (%) | F1-Score (%) | Accuracy (%) |
|---|---|---|---|---|---|
| Resnet50 | No | 67.91 | 66.70 | 66.95 | 76.02 |
|  | Yes | 82.45 | 81.83 | 81.69 | 81.83 |
| VGG16 | No | 35.77 | 34.77 | 32.51 | 55.33 |
|  | Yes | 83.78 | 83.21 | 83.17 | 83.21 |
| MobileNetv2 | No | 56.66 | 52.75 | 53.42 | 65.22 |
|  | Yes | 73.51 | 72.02 | 71.52 | 72.02 |

*4.2. Transfer Learning Evaluation*

To train the three models (ResNet, VGGNet and MobileNet) using the IP_CitrusPests dataset, pre-trained weights from the ImageNet dataset were utilized in this experiment. Figure 7 illustrates the loss curves for the training and validation sets of the three networks,

both before and after transfer learning. From the Figure, it is evident that the training and validation loss curve of the classification network exhibit a sharp decrease with an increase in epoch in the absence of transfer learning. On the contrary, the decreasing trend of the training and validation loss curve of the classification model with the introduction of transfer learning is more moderate. It is shown that transfer learning can significantly reduce the occurrence of the overfitting phenomenon. Even in the context of complex and diverse environments, the ResNet50, VGG16 and MobileNetV2 can still get lower loss values and better generalization ability after introducing transfer learning.

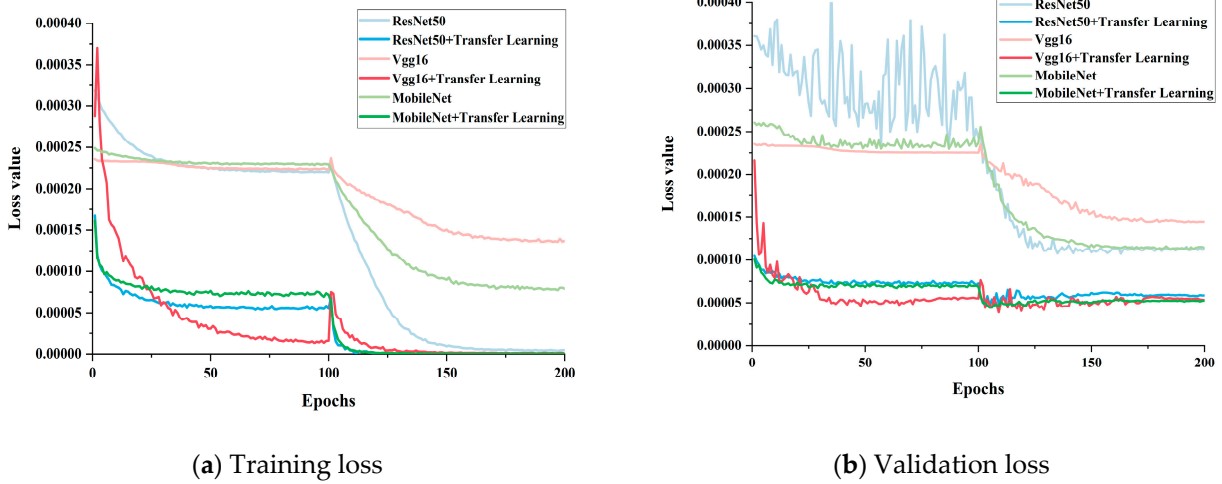

(**a**) Training loss  (**b**) Validation loss

**Figure 7.** The loss curves for the training and validation sets of the ResNet, VGGNet and MobileNet, both before and after transfer learning. (**a**) Is the loss curve for the training set, (**b**) is the loss curve for the validation set.

To more clearly visualize the different changes in the network models before and after the incorporation of transfer learning, Figure 8 shows the visual feature maps of the shallow networks of ResNet50, VGG16 and MobileNetV2 with sample images from the *Adris Tyrannus* class in the IP_CitrusPests. It is evident from the figure that the shallow neural network before using transfer learning simply changes the geometric color of the sample images, while the shallow network after using transfer learning can well extract generic information such as edges, contours and textures of the pests in the sample images, clustering the geometric shapes of the pest samples, and the features learned by these shallow networks are basically generic across all sample images. As the convolutional layer deepens, the clustering only becomes more and more specific, so the knowledge learned by the shallow network can be transferred to different datasets for further training.

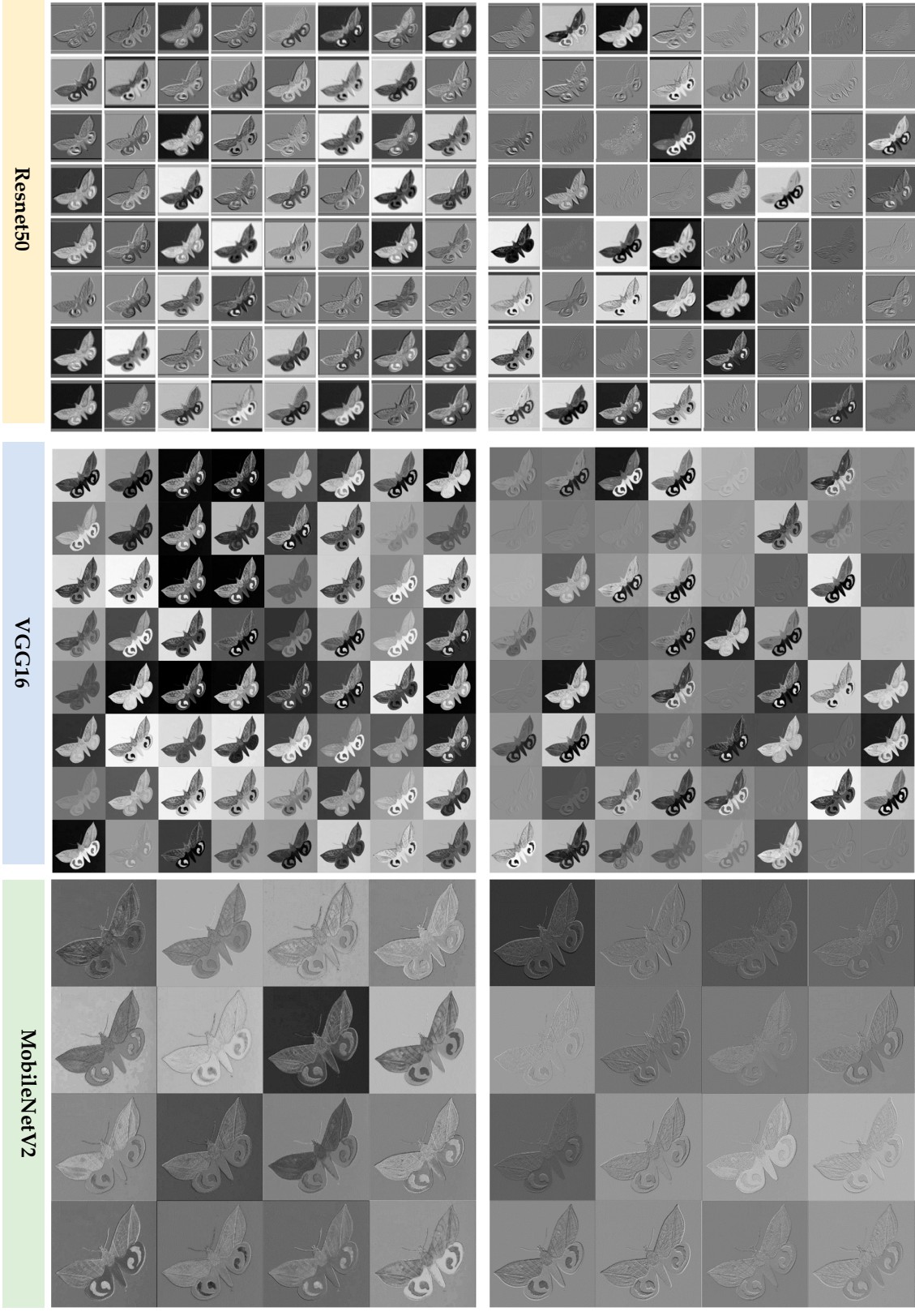

**No transfer learning**   **Transfer learning**

**Figure 8.** Visualization feature map of the shallow network of ResNet50, VGG16 and MobilenetV2 before and after using transfer learning.

Table 6 provides a comparative analysis of precision, recall, F1-score and accuracy of the IP_CitrusPests across ResNet50, VGG16 and MobileNetV2 before and after the integration of transfer learning. The table highlights an improvement in all evaluation metrics for all models following the integration of transfer learning. Specifically, the average accuracy increased from 65.52% to 89.68%, indicating an improvement of 25.16%. In particular, for a larger and more parameterized model such as the VGG16, the accuracy before the integration of transfer learning was only 55.33%, while the accuracy reached 90.24% after the integration of pre-trained ImageNet weights, indicating a direct improvement of 34.91%; it demonstrates that transfer learning exerts a considerable influence on the accuracy of the VGG16 impact. The classification accuracy of ResNet50 and MobileNetV2 classification models also improved by 13.51% and 24.05%, respectively. The results intuitively showed that the integration of transfer learning can lead to substantial performance improvements for all three models in all categories.

**Table 6.** The comparison of precision, recall, F1-score and accuracy of the IP_CitrusPests on ResNet50, VGG16 and MobileNetV2 before and after the integration of transfer learning.

| Model | Transfer Learning | Precision (%) | Recall (%) | F1-Score (%) | Accuracy (%) |
|---|---|---|---|---|---|
| Resnet50 | No | 67.91 | 66.70 | 66.95 | 76.02 |
| | Yes | 84.37 | 81.89 | 82.79 | 89.53 |
| VGG16 | No | 35.77 | 34.77 | 32.51 | 55.33 |
| | Yes | 85.69 | 83.16 | 84.01 | 90.24 |
| MobileNetv2 | No | 56.66 | 52.75 | 53.42 | 65.22 |
| | Yes | 83.84 | 81.93 | 82.51 | 89.27 |

### 4.3. Attention Module Assessment

In this study, effective attention mechanisms were introduced for the characteristics of ResNet50, VGG16 and MobileNetV2. Firstly, the RGA module was embedded after each residual block of the ResNet50 network model. Secondly, the ECA module was embedded in the VGG16. Thirdly, the CA attention mechanism was embedded before the last bottleneck of the MobileNetV2 (see Section 3.4 for details). Table 7 shows the comparison of each evaluation metric before and after adding the attention mechanism to ResNet50, VGG16 and MobileNetV2 on the IP_CitrusPests. Based on the table, it is evident that the precision, recall, F1-Score and accuracy metrics experienced improvement upon the integration of the RGA, ECA and CA on ResNet50, VGG16 and MobileNetV2, respectively. The average accuracy of the three networks was also enhanced by 2.76%. In addition, ResNet50 and MobileNet also improved by 1.29% and 1.30%, respectively, this is because all three CNNs focused more on the important features of the pest itself after adding the attention mechanism module.

**Table 7.** The comparison of each evaluation metric before and after adding the attention mechanism to ResNet50, VGG16 and MobileNetV2 on the IP_CitrusPests dataset.

| Model | Attention | Precision (%) | Recall (%) | F1-Score (%) | Accuracy (%) |
|---|---|---|---|---|---|
| Resnet50 | No | 67.91 | 66.70 | 66.95 | 76.02 |
| | Yes | 69.63 | 69.99 | 69.18 | 77.31 |
| VGG16 | No | 35.77 | 34.77 | 32.51 | 55.33 |
| | Yes | 57.47 | 45.30 | 47.96 | 61.02 |
| MobileNetv2 | No | 56.66 | 52.75 | 53.42 | 65.22 |
| | Yes | 58.62 | 56.07 | 56.14 | 66.52 |

To visualize the areas that the networks focus more on before and after adding the attention mechanism, we used Gradient-weighted Class Activation Mapping (Grad-CAM) [43]

to visualize the feature areas that the classification network pays more attention to in the form of a heat map (red regions). It is obvious from Figure 9 that the red areas in the heat map in the second row (classification network without attention mechanism module added) are mixed with many complex background areas, as in the heat map of VGG16 (second plot in the second row) where the red areas are distributed over citrus and other backgrounds rather than the pest itself. However, the red regions in the heat map in the third row all accurately cover the pest itself, with resnet50 performing the best, successfully learning features along the outline of the pest's shape. It is demonstrated that with the addition of the attention mechanism, the ResNet50, VGG16 and MobileNetV2 enable the model to learn important features of the pest more quickly in complex backgrounds.

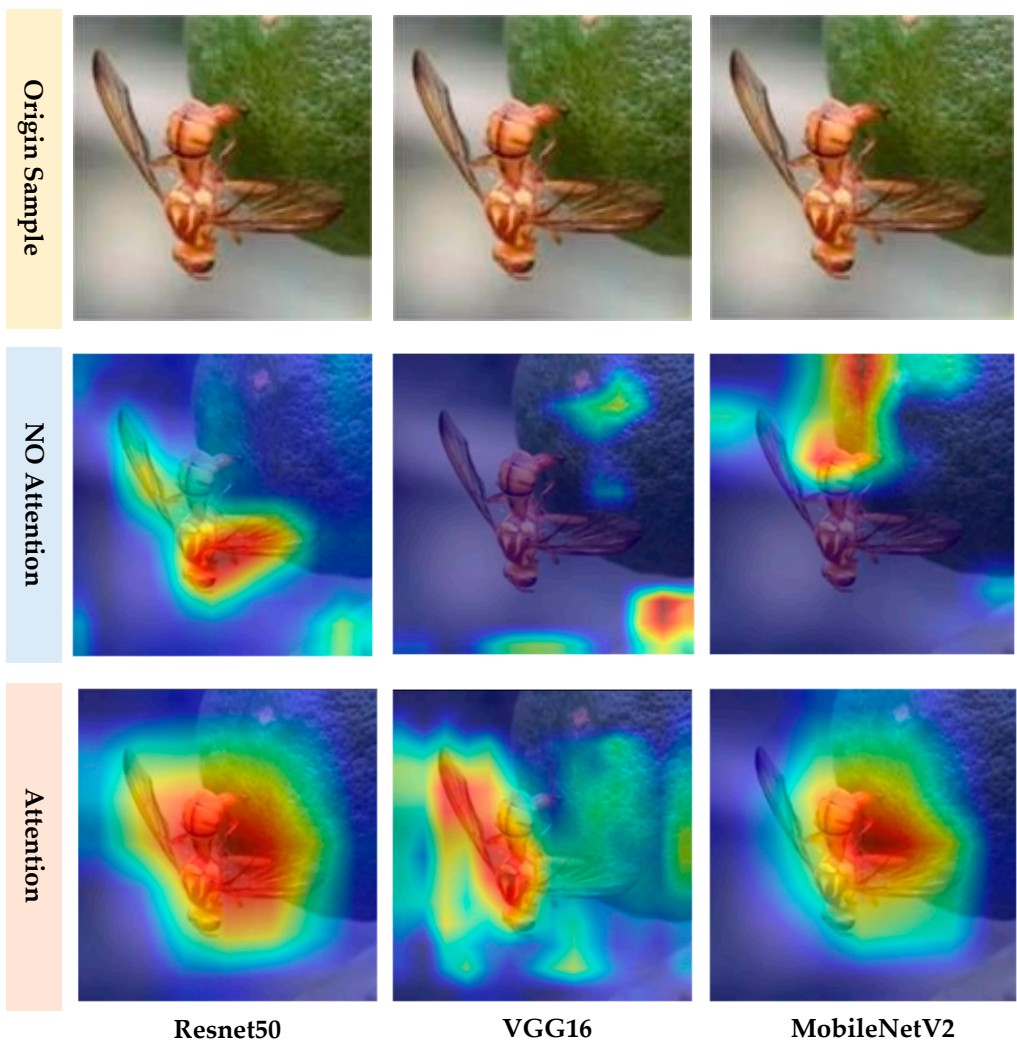

**Figure 9.** Heat map comparison of ResNet50,VGG16 and MobileNetV2 before and after adding attention mechanism.

*4.4. Total Method Evaluation*

This experiments also investigated the classification effects of AR-GAN data augmentation, transfer learning and attention mechanism (Resnet50_RGA, VGG16_ECA and MobileNetV2_CA) on the IP_CitrusPests together with the ResNet50, VGG16 and MobileNetV2, as shown in Table 8. We can see that the accuracy of Resnet50 combined with AR-GAN data augmentation, transfer learning and the RGA can reach 93.81%, which represented a substantial improvement of 17.79% when compared to the primitive Resnet50's accuracy of 76.02%. The VGG16 was able to achieve 93.65% classification accuracy after combining the three methods, a significant improvement of 38.32% compared to 55.33% of

the original network. MobileNetV2 achieved an accuracy of 93.48% after combining the three methods, a great improvement of 28.26% in comparison to the original network's accuracy score of 65.22%. In addition, the table also showed that after introducing AR-GAN data augmentation, transfer learning and attention model simultaneously on the three classification networks, precision, recall and F1-Score were all significantly improved, with VGG16 showing significant changes in the values of each evaluation metric by 57.66%, 58.88% and 61.12%, respectively. Overall, ResNet50 achieved the best classification performance combining the three methods on the IP_CitrusPests dataset, with precision at 94.06%, recall at 93.81%, F1-Score at 93.76% and accuracy at 93.81%. The superiority of the three methods on three networks was demonstrated for the IP_CitrusPests.

**Table 8.** The comparison of each evaluation metric before and after adding the three methods (ARGAN data argument, transfer learning, and attention mechanism) to ResNet50, VGG16 and MobileNetV2 on IP_CitrusPests.

| Method | Precision (%) | Recall (%) | F1-Score (%) | Accuracy (%) |
|---|---|---|---|---|
| Resnet50 | 67.91 | 66.70 | 66.95 | 76.02 |
| Resnet50 + AR-GAN + Transfer Learning + RGA | 94.06 | 93.81 | 93.76 | 93.81 |
| VGG16 | 35.77 | 34.77 | 32.51 | 55.33 |
| VGG16 + AR-GAN + Transfer Learning + ECA | 93.76 | 93.65 | 93.63 | 93.65 |
| MobileNetV2 | 56.66 | 52.75 | 53.42 | 65.22 |
| MobileNetV2 + AR-GAN + Transfer Learning + CA | 93.65 | 93.48 | 93.46 | 93.48 |

*4.5. Comparison with State-of-the-Art Models*

Table 9 shows the comparison of the qualitative and quantitative features of the proposed method with the current work. It can be seen that the proposed method (IP_CitrusPests dataset after AR-GAN data augmentation, combined with transfer learning and attention mechanisms trained on the classification model) outperforms other current work on accurate pest classification with a final accuracy of 93.65%.

**Table 9.** Qulitive and quantitative comparison between our work and existing work.

| Paper | Dataset | Classes | Method | Accuracy (%) |
|---|---|---|---|---|
| [29] | IP102 | 102 | CNN | 49.50 |
| [11] | IP102 | 102 | CNN + Transfer Learning + Traditional Data Augmentation | 57.08 |
| [44] | Citrus Pests and Diseases | 18 | Weakly DenseNet + Transfer Learning + Traditional Data Augmentation | 93.33 |
| [27] | IP102 | 102 | CNN + Attention | 73.29 |
| [28] | Citrus Pests | 2 | Improved cascade R-CNN + Attention | 88.78 |
| [45] | Rice Pests | 14 | CNN + Transfer Learning + ARGAN | 86.96 |
| Ours | IP_CitrusPests | 14 | CNN + Transfer Learning + ARGAN + Attention | 93.65 |

## 5. Discussion

In the complex citrus growing environment, factors such as climate change and the small size of pests make it difficult to collect sample data on citrus pests. In this study, a large-scale dataset, IP_CitrusPests, was compiled, which included data of 14 types of citrus pest samples with a total of 5182 images (IP_CitrusPests dataset was open sourced in https://github.com/ValenciaJXQ/IP_CitrusPests.git, accessed on 12 Mar 2023). However, the IP_CitrusPests dataset has a high imbalance ratio (see Section 3.1). Therefore, we introduced AR-GAN data augmentation as a way to expand the pest categories with a small sample size, so that the quantity of each category in IP_CitrusPests was increased

to 1500. The experimental data suggested that the average accuracy of pests after data augmentation was improved by 13.5% on the ResNet50, VGG16 and MobileNetV2 (see Section 4.1). In particular, pests such as *Parlatoria zizyphus Lucus*, *Phyllocoptes oleiverus ashmead* and *Toxoptera citricidus* with small initial data sample sizes improved the average accuracy by 46.57%, 51.83% and 54.81% on the networks, respectively, which fully proved that AR-GAN data augmentation could solve the problem of highly unbalanced data sets to some extent.

Transfer learning can effectively solve this problem when the training dataset is too small and the target domain data is difficult to collect. Therefore, in this study, a parameter-based isomorphic transfer learning method was used to train and update the model after replacing the new layer on the new dataset, IP_CitrusPests, according to the case that the dataset processing domains have the same feature space. Based on the experimental findings, the average accuracy of pests after transfer learning was 89.68% on the ResNet50, VGG16 and MobileNetV2, which was a substantial improvement of 24.16% compared with the average classification accuracy of 65.52% for the network trained from scratch (see Section 4.2). It was demonstrated that the introduction of transfer learning on the ResNet50, VGG16 and MobileNet can effectively improve the generalization ability of the models while improving the model training speed and saving computational resources.

To better localize to the pest itself in a complex background environment, we added the appropriate attention mechanism to each of the three classification network models according to the variability of the networks (see Section 3.4). Firstly, in this paper, the RGA module, which can consider global information in spatial and channel dimensions, was added after each residual block of ResNet50. Secondly, the ECA module, which can extract inter-channel dependencies, was added to the deep network of VGG16. Finally, the CA module, which can consider the relationship between space, channel and position simultaneously, was embedded before the last inverse residual module of the network. The average accuracy of pests on the models after adding the attention mechanism was improved by 2.76% (see Section 4.3). The experimental outcomes indicated that incorporating the attention mechanism into the CNNs resulted in more efficient targeting of the sample pests. As a result, the features extracted by the models were more dependent on the pests themselves rather than the intricate environmental factors surrounding them.

## 6. Conclusions

We collected a citrus pest dataset, IP_CitrusPests, for accurate identification of citrus pests, which included 5182 sample data with a total of 14 major pest classes. To address the issue of imbalanced data distribution across the IP_CitrusPests categories, we introduced AR-GAN data augmentation to expand each class of pests to 1500. To improve the model speed and save computational resources, we used transfer learning based on parameter fine-tuning to train on three common classification models (ResNet50, VGG16 and MobileNetV2) and embed the attention module to the most appropriate position of the network (Resnet50_RGA, VGG16_ECA and MobileNetV2_CA) to make it emphasize the characteristics of the pest samples themselves. According to the experimental findings, the average accuracy of the IP_CitrusPests after AR-GAN on the three networks combining transfer learning and attention mechanism reached 93.64%, which was 28.12% higher than the initial average accuracy of 65.52% on the three models. It proved that the dataset enhanced by AR-GAN on the CNNs combining transfer learning and attention module can significantly save the training cost and time, effectively improve the accuracy and essentially solve the problem of the unbalanced number of samples among data classes.

We used deep learning technology to identify citrus pests efficiently, quickly, accurately and cost-effectively, and then used green control methods such as changing field microclimate and releasing natural enemy insects to achieve early control of citrus pests, which not only ensures safe crop growth but also protects the environment. In the future, we hope to collect more images of high-quality data samples containing complex backgrounds

based on our existing data resources and continue to explore the precise localization and identification of citrus pests.

**Author Contributions:** Conceptualization, Y.N. and J.M.; methodology, J.M.; software, Z.L. and J.M.; validation, X.J. (Xueqin Jiang), X.J. (Xinyu Jia). and Y.N.; formal analysis, Y.W and X.J. (Xueqin Jiang); investigation, Z.L. and X.J. (Xinyu Jia); resources, Z.L. and Y.W; data curation, Z.L. and X.J. (Xinyu Jia); writing—original draft preparation, J.M. and Z.L.; writing—review and editing, Y.N. and J.M.; visualization, X.J. (Xueqin Jiang) and Y.N.; supervision, X.J. (Xinyu Jia),Y.W. and Y.W; project administration, X.J. (Xueqin Jiang) and Y.W.; funding acquisition, J.M. and X.J. (Xinyu Jia) All authors have read and agreed to the published version of the manuscript.

**Funding:** This research was funded by Research and application of key technologies for intelligent spraying based on machine vision (key technology research project) of Sichuan Provincial Department of Science and Technology, grant number 22ZDYF0095.

**Data Availability Statement:** The data in this study are available upon request from the corresponding author.

**Acknowledgments:** Thanks to Jingfan Chen and Runxing Chang for their help.

**Conflicts of Interest:** The authors declare no conflict of interest.

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
