# Peer review of "Application of Deep Learning in Image Recognition of Citrus Pests"

_agriculture, doi:10.3390/agriculture13051023_

Round 1

Reviewer 1 Report

Please check attachment 

Reviewer 2 Report

In this articles authors developed a dataset containing 5182 pest images with 14 categories of citrus pest. Afterwards, they used Attentive Recurrent Generative Adversarial Networks (AR-GAN) data augmentation technique to expand the dataset to 21000 images. Then, they built Visual Geometry Group Network (VGG), Residual Neural Network (ResNet) and MoblieNet citrus pest recognition models by using transfer learning, and finally, introduced appropriate attention mechanism according to the model characteristics respectively, so that the three network models pay more attention to the deep features of the pests themselves in complex real environments. The overall idea is interesting to overcome the issue of balance class size. The paper is well written, however, contribution of the paper in the last paragraph of the introduction can be improved further.  There should be a separate section for the literature review.

Reviewer 3 Report

The authors collected a large citrus pest dataset, introduced AR-GAN data augmentation to expand each class of pests to 1500, used transfer learning method to train three common classification network models, and embed the attention module to the most appropriate position of the network. It is an interesting study. However, this manuscript should be improved in the following points:

1)     The method did not provide enough information because it cannot be repeated by other researchers based on the provided information (please see my detailed comments).

2)     Too much information is repeated (please see my detailed comments).

3)     The English statements should be improved (please see my detailed comments). It should be in past tense when you described the method used by you.

Detailed comments

L62 what is the difference between four public datasets and four common datasets. Rephrase this statement.

L115 to 118 you already mentioned in the introduction.

L124: please specify the more complex environments and backgrounds.

L146. Rephrase the title of the table 1 (delete “this is the”).

L148. Delete “refer to”. In the text, the scientific name of the insects should be italic.

By the way, I do not think Fig.1 is needed. Also, Fig. 2 repeated the information presented in Table 1. You should delete one.

L157. The statement is too long, rephrase it.

L180. You need to tell reader how the generator works.

2.3 You provide enough information on how other researchers did. You need to tell readers how you did.  No body can repeat your study based on your description.

L282: it should be: Precision is the proportion of samples with positive prediction results of the true cases.

L303 to 311. You already mentioned it in the methodology.

Fig. 7 is not needed because it repeated Fig. 2, and you increased the number to 1500 regardless of the original dataset.

Table 4.  Put the following information in the footnote:  AT refers to Adris Tyrannus; As refers to Aleurocanthus 344 spiniferus; Bt refers to Bactrocera tsuneonis; Cr refers to Ceroplastes rubens; Ca refers to 345 Chrysomphalus aonidum; DdH refers to Dacus dorsalis Hendel; PcM refers to Panonchus citri 346 McGregor; Px refers to Papilio xuthus; PzL refers to Parlatoria zizyphus Lucus; Poa refers to Phyl-347 locoptes oleiverus ashmead; Pl refers to Prodenia litura; Tc refers to Toxoptera citricidus; Ta refers 348 to Toxoptera aurantii and Uy refers to Unaspis yanonensis.

Too much information in the discussion repeated the information presented in the introduction and methods.

Round 2

Reviewer 1 Report

Author has addressed all the comments 

Reviewer 3 Report

No further comment.